# Antipodes of Label Differential Privacy: PATE and ALIBI

**Mani Malek**[*]    **Ilya Mironov**[*]    **Karthik Prasad**[*]    **Igor Shilov**[*]    **Florian Tramèr**[†]

## Abstract

We consider the privacy-preserving machine learning (ML) setting where the trained model must satisfy differential privacy (DP) with respect to the *labels* of the training examples. We propose two novel approaches based on, respectively, the Laplace mechanism and the PATE framework, and demonstrate their effectiveness on standard benchmarks.

While recent work by Ghazi et al. proposed Label DP schemes based on a randomized response mechanism, we argue that additive Laplace noise coupled with Bayesian inference (ALIBI) is a better fit for typical ML tasks. Moreover, we show how to achieve very strong privacy levels in some regimes, with our adaptation of the PATE framework that builds on recent advances in semi-supervised learning.

We complement theoretical analysis of our algorithms' privacy guarantees with empirical evaluation of their memorization properties. Our evaluation suggests that comparing different algorithms according to their *provable* DP guarantees can be misleading and favor a less private algorithm with a tighter analysis.

Code for implementation of algorithms and memorization attacks is available from `https://github.com/facebookresearch/label_dp_antipodes`.

## 1 Introduction

Sophisticated machine learning models perform well on their target tasks despite of—or thanks to—their predilection for data memorization [39, 2, 17]. When such models are trained on non-public inputs, privacy concerns become paramount [31, 6, 7], which motivates the actively developing area of privacy-preserving machine learning.

With some notable exceptions, which we discuss in the related works section, existing research has focused on an "all or nothing" privacy definition, where all of the training data, i.e., both *features* and *labels*, is considered private information. While this goal is appropriate for many applications, there are several important scenarios where *label-only* privacy is the right solution concept.

A prominent example of label-only privacy is in online advertising, where the goal is to predict conversion of an ad impression (the label) given a user's profile and the spot's context (the features). The features are available to the advertising network, which trains the model, while the labels (data on past conversion events) are visible only to the advertiser. More generally, any two-party setting where data is vertically split between public inputs and (sensitive) outcomes is a candidate for training with label-only privacy.

In this work we adopt the definition of differential privacy as our primary notion of privacy. The definition, introduced in the seminal work by Dwork et al. [13], satisfies several important properties that make it well-suited for applications: composability, robustness to auxiliary information, preservation under post-processing, and graceful degradation in the presence of correlated inputs (group privacy). Following Ghazi et al. [18], we refer to the setting of label-only differential privacy as Label DP.

---

[*]`{manimalek,imironov,krp,shilov}@fb.com`, Facebook AI.

[†]`tramer@cs.stanford.edu`, Stanford University.

35th Conference on Neural Information Processing Systems (NeurIPS 2021).

**Our contributions.** We explore two approaches—with very different characteristics—toward Label DP. These approaches and our methodology for estimating empirical privacy loss via label memorization are briefly reviewed below:

- *PATE*: We adapt the PATE framework (Private Aggregation of Teacher Ensembles) of Papernot et al. [28] to the Label DP setting by observing that PATE's central assumption—availability of a public unlabeled dataset—can be satisfied for free. Indeed, a public unlabeled dataset is simply the private dataset with the labels *removed*! By instantiating PATE with a state-of-the-art technique for semi-supervised learning, we demonstrate an excellent tradeoff between empirical privacy loss and accuracy.
- *ALIBI (Additive Laplace with Iterative Bayesian Inference)*: We propose applying additive Laplace noise to a one-hot encoding of the label. Since differential privacy is preserved under post-processing, we can de-noise the mechanism's output by performing Bayesian inference using the prior provided by the model itself, doing so continuously during the training process. ALIBI improves, particularly in a high-privacy regime, Ghazi et al.'s approach based on randomized response [18].
- *Empirical privacy loss.* We describe a memorization attack targeted at the Label DP scenario that efficiently extracts lower bounds for the privacy parameter that, in some cases, come within a factor of 2.5 from the theoretical, worst-case upper bounds against practically relevant, large-scale models. For comparison, recent, most advanced black-box attacks on DP-SGD have approximately a $10\times$ gap between the upper and lower bounds [20, 26].

Both algorithms, PATE and ALIBI, come with rigorous privacy analyses that, for any fixed setting of parameters, result in an $(\varepsilon, \delta)$-DP bound. We emphasize that these bounds are just that—they are upper limits on an attacker's ability to breach privacy (in a precise sense) on worst-case inputs. Even though these bounds can be pessimistic, intuitively, a smaller $\varepsilon$ for the same $\delta$ corresponds to a more private instantiation of a mechanism, and indeed this is likely the case within an algorithmic family.

Can the privacy upper bounds be used to compare diverse mechanisms with widely dissimilar analyses, such as ours? We argue that focusing on the upper bounds alone may be misleading to the point of favoring a less private algorithm with tighter analysis over a more private one whose privacy analysis is less developed. For a uniform perspective, we estimate the *empirical* privacy loss of all trained models by evaluating the performance of a black-box membership inference attack [31, 20, 26].

Our attack directly measures memorization of deliberately mislabeled examples, or *canaries* [6]. Following prior work [26, 20], we use the success rate of our attack to compute a lower bound on the level $\varepsilon$ of label-DP achieved by the training algorithm. To avoid the prohibitive cost of retraining a model for each canary, we propose and analyze a heuristic that consists in inserting multiple canaries into one model, and measuring an attacker's success rate in guessing each canary's label. In some settings, we find that the empirical privacy of PATE (as measured by our attack) is significantly stronger than that of ALIBI, even though the provable DP bound for PATE is much weaker.

Finally, to characterize setups where label-only privacy may be applicable, it is helpful to distinguish between *inference* tasks, where the label is fully determined by the features, and *prediction*, where there is some uncertainty as to the eventual outcome, despite (hopefully) some predictive value of the feature set. We are mostly concerned with the latter scenario, where the outcome is not evident from the features, or protecting choices made by individuals may be mandated or desirable (for connection between privacy and self-determination see Schwartz [30]). While we do use public data sets (CIFAR-10 and CIFAR-100) in which it is possible to "infer" the label, we do so only to evaluate our approaches on standard benchmarks. We completely remove the triviality of *inference* by adding mislabeled canaries in our attack, thereby introducing a measure of non-determinism.

## 2 Notation and Definitions

Throughout the paper we consider standard supervised learning problems. The inputs are $(x, y)$ pairs where $x \in X$ are the *features* and $y \in Y$ are the *labels*. (If some labels are absent, the problem becomes semi-supervised learning.) The cardinality of the set $Y$ is the number of *classes* $C$. The task is to learn a model $M$ parameterized with weights $W$ that predicts $y$ given $x$. To this end the training algorithm has access to a training set of $n$ samples $\{(x_i, y_i)\}_n$.

Differential privacy due to Dwork et al. [13] is a rigorous, quantifiable notion of individual privacy for statistical algorithms. (See Dwork and Roth [14] and Vadhan [33] for book-length introductions.)

**Definition 2.1** (Differential Privacy). *We say that a randomized mechanism $\mathcal{M} \colon U \to R$ satisfies $(\varepsilon, \delta)$-differential privacy (DP) if for all adjacent inputs $D, D' \in U$ that differ in contributions of a single sample, the following holds:*

$$\forall S \subset R \quad \Pr[\mathcal{M}(D) \in S] \leq e^\varepsilon \cdot \Pr[\mathcal{M}(D') \in S] + \delta.$$

*If $\delta = 0$, we refer to it as $\varepsilon$-DP.*

The notion of adjacency between databases $D$ and $D'$ is domain- and application-dependent. To instantiate the Label DP setting, we say that $D$ and $D'$ are adjacent if they consist of the same number of examples $\{(x_i, y_i)\}_n$ and are identical except for a difference in a single label at index $i^*$: $(x_{i^*}, y_{i^*}) \in D$ and $(x_{i^*}, y'_{i^*}) \in D'$.

## 3 Background and related work

**DP and machine learning.** Differential privacy aligns extremely well with the goal of machine learning, which is to produce valid models or hypotheses on the basis of observed data. In fact, the relationship can be made precise. An important metric in ML is *generalization error*, which measures the model's performance on previously unseen samples. Differentially private mechanisms provably generalize [11]. For an introduction to some important early results on connections between differential privacy and learning theory see Dwork and Roth [14, Chapter 11].

**Randomized response (RR).** As a disclosure avoidance methodology, randomized response (RR) [36] precedes the advent of differential privacy. In RR's basic form, with some probability the sensitive input is replaced by a sample drawn uniformly from its entire domain. RR can be implemented at the source, making it an essential building block in achieving Local DP [16].

More generally, RR is an instance of the *input perturbation* technique, which leaves the ML training algorithm intact and achieves differential privacy via input randomization. Input perturbations methods are particularly friendly to implementers as they require minimal changes to existing algorithms, treating them as black boxes.

**Additive noise mechanisms.** Another important class of differentially private mechanisms are *additive noise* methods. Two canonical examples are additive Laplace [13] and additive Gaussian [12] that can be applied to vector-valued functions. Their noise distributions are calibrated to the function's *sensitivity*—the maximal distance between the function's outputs on adjacent inputs ($D$ and $D'$ in Definition 2.1) under the $\ell_1$ or $\ell_2$ metrics for the Laplace and Gaussian mechanisms respectively.

**PATE.** Private Aggregation of Teacher Ensembles (PATE) [28, 29] is a framework based on private knowledge aggregation of an ensemble model and knowledge transfer. PATE trains an ensemble of "teachers" on disjoint subsets of the private dataset. The ensemble's knowledge is then transferred to a "student" model via differentially private aggregation of the teachers' votes on samples from an unlabeled public dataset. Only the student model is released as the output of the training, as it accesses sensitive data via a privacy-preserving interface.

Since PATE consumes the privacy budget each time the student queries the teachers, the training algorithm must be optimized to minimize the number of queries. To this end, Papernot et al. explore the use of semi-supervised learning techniques, most notably GANs. Recent advances in semi-supervised learning have been shown to boost the performance of PATE on standard benchmarks [5].

**FixMatch.** FixMatch [32] is a state-of-the-art semi-supervised learning algorithm, which achieves high accuracy on benchmark image datasets with very few labeled examples. It is based on the concepts of pseudo-labeling [23] and consistency regularization [3]: FixMatch uses a partially trained model to generate labels on weakly-augmented unlabeled data, and then, if the prediction is confident enough, uses the output as a label for the strongly-augmented version of the same data point.

**Label-only privacy.** The setting of label-only DP was formally introduced by Chaudhuri and Hsu [8], who proved lower bounds on sample complexity for private PAC-learners. Beimel et al. [4] demonstrated that the sample complexity of Label DP matches that of non-private PAC learning (ignoring constants, dependencies on the privacy parameters, and computational efficiency). Wang and Xu [34] considered the linear regression problem in the local, label-only privacy setting.

Most recently, Ghazi et al. [18] considered label-only privacy for deep learning. They propose a Label Privacy Multi-Stage Training (LP-MST) algorithm, which we revisit in Section 5. The algorithm advances in stages (Ghazi et al. evaluate it up to four), training the model on disjoint partitions of the dataset. In each stage, the private labels are protected using a randomized response scheme.

The most innovative part of LP-MST happens at the point of transition between stages. Labels of the training examples of the next partition are perturbed using randomized response. The (soft) output of the previous stage's model is interpreted as a prior, which is used to choose parameters for the randomized response algorithm *separately for each example*. For instance, if the current model assigns high probabilities to labels 1–10 and low probabilities to labels 11–100, the randomized response will be constrained to outputting a label within the set 1–10.

We build on the insights of Ghazi et al. by making two observations. First, training algorithms used for deep learning (such as SGD and its variants) can naturally handle soft labels (i.e., probability distributions over the label space); there is thus no need to force hard labels by sampling from the posterior. Second, by applying additive noise, we can repeatedly recompute the posterior distribution as better priors become available *without* consuming privacy budget. By following the "once and done" approach to label perturbation, we forgo the need to split training into disjoint stages.

To summarize, Ghazi et al. fix the prior at the beginning of the stage, choose parameters of the randomized response algorithm on the basis of this prior, and update the model by feeding it the perturbed label. We flip the order of these operations by applying a DP mechanism (additive Laplace) first, and then repeatedly use Bayesian inference by combining the prior that changes with each model update and the observables that do not.

To our knowledge, we are the first to explore connections between PATE and Label DP.

## 4   PATE with Semi-Supervised Learning

The PATE framework [28] splits the learning procedure into two stages. First, $T$ teachers are trained on disjoint partitions of the private dataset. Second, a student acquires labels for training its own model by querying the teacher ensemble on samples drawn from a public (non-sensitive) unlabeled dataset. Differential privacy is enforced at the interface between the student and the teachers, by means of a private aggregation of the teachers' votes.

The critical observation underlying the PATE privacy analysis is that a private sample taints at most one teacher due to disjointedness of the teachers' training sets. The voting mechanism bounds the maximal impact of a single teacher's votes across all student queries by injecting noise in each voting instance. (We use the Confident Gaussian aggregation mechanism and its analysis from [29].)

The Label DP setting allows for unrestricted use of the features. We leverage this capability for training both teachers and the student:

- The unlabeled examples on which the student queries the teacher ensemble are random samples from the dataset *without* labels.
- All models—the teachers and the student—are trained with the FixMatch algorithm that uses (few) labeled examples and has access to the entirety of the original dataset as the supplementary unlabeled data.

Algorithms 1 and 2 adapt PATE to the Label DP setting. To draw a distinction between the PATE framework, which is generic and black-box, and its instantiation with the FixMatch algorithm for semi-supervised learning, we refer to the latter as PATE-FM.

The PATE framework does not require the student and the teacher models to have identical architecture, or share hyperparameters. In our experience, however, the optimal privacy/accuracy trade-offs force very similar choices on these models.

The selection of the number of teachers, $T$, balances two competing objectives. On the one hand, each teacher has access to $n/T$ labeled and $n$ unlabeled examples, which means that a smaller $T$ corresponds to higher accuracy teacher models (due to more labeled examples per teacher). On the other hand, a larger $T$ allows for a higher level of noise and lower privacy budget per query.

A similar dynamic controls $K$, the number of labels requested by the student. A higher number allows for more accurate training but inflates the privacy budget.

---

**Algorithm 1:** `PATE-FM.TrainTeacherEnsemble`

---

**Input**: Dataset $D \leftarrow \{(x_1, y_1), \ldots, (x_n, y_n)\}$, number of teachers $T$, training procedure `FixMatch`($labeled\_data, unlabeled\_data$)

Define the unlabeled part of $D$: $D^- \leftarrow \{x_1, \ldots, x_n\}$
Partition $D$ into $T$ equal-sized disjoint subsets $D^{(1)}, \ldots, D^{(T)}$
**for** $i \leftarrow 1$ **to** $T$ **do**
   |  $teacher_i \leftarrow$ `FixMatch`($D^{(i)}, D^-$)
**end**
**Output**: Model ensemble $\{teacher_1, \ldots, teacher_T\}$

---

---

**Algorithm 2:** `PATE-FM.TrainStudent`

---

**Input**: Dataset $D = \{(x_1, y_1), \ldots, (x_n, y_n)\}$, number of label classes $C$, number of teachers $T$, training procedure `FixMatch`($labeled\_data, unlabeled\_data$), number of student samples $K$, noise parameters $\sigma_1, \sigma_2$, voting threshold $\tau$

Define the unlabeled part of $D$: $D^- \leftarrow \{x_1, \ldots, x_n\}$
Initialize student dataset $D_S \leftarrow \emptyset$
$\{teacher_1, \ldots, teacher_T\} \leftarrow$ `TrainTeacherEnsemble`($D, T$)
**while** $|D_S| < K$ **do**
   |  $x \leftarrow$ random sample from $D^-$
   |  **for** $i \leftarrow 1$ **to** $C$ **do**   // tallying up the votes
   |    |  $v_i \leftarrow |\{j : teacher_j(x) = i\}|$
   |  **end**
   |  **if** $\max_i\{v_i + \mathcal{N}(0, \sigma_1^2)\} \geq \tau$ **then**
   |    |  $y \leftarrow \text{argmax}_i\{v_i + \mathcal{N}(0, \sigma_2^2)\}$
   |    |  $D_S \leftarrow D_S \cup \{(x, y)\}$
   |  **end**
**end**
$student \leftarrow$ `FixMatch`($D_S, D^-$)
**Output**: Model $student$

---

Boosting teachers' accuracy for a given number of labels and minimizing the number of student queries and their privacy costs are key to producing high utility models with strong privacy guarantees. To this end, we use the state-of-the-art algorithm for semi-supervised learning FixMatch [32].

The FixMatch algorithm's primary domain is image classification: in addition to labelled and unlabelled examples, it assumes access to a pair of weak and strong data augmentation algorithms. The loss function of FixMatch is a weighted sum of two losses that enforce consistency of the model's prediction (i) between labeled examples and their weakly augmented versions, and (ii) between weakly and strongly augmented unlabeled images (restricted to high-confidence instances).

## 5 ALIBI: Additive Laplace with Iterative Bayesian Inference

We describe the general approach of Soft Randomized Response with post-processing and its concrete instantiation, ALIBI.

### 5.1 Soft Randomized Response

Consider the application of *Randomized Response (RR)* to the setting of Label DP. In its standard form, the random perturbation maps labels to labels, i.e., the label either retains its value with probability $p$ or assumes a random value with probability $1 - p$.

We deviate from RR by replacing label randomization with a differentially private mechanism applied to a one-hot encoding of the training sample label; we refer to this mechanism as Soft Randomized Response (Soft-RR). By retaining more information (without significantly impacting privacy analysis), we support several possible post-processing algorithms (Section 5.2). Additionally, we argue that Soft-RR performs better than RR in practice since it does not force wrong hard labels.

**Algorithm 3:** Additive Laplace with Iterative Bayesian Inference, `ALIBI`

---

**Input**: Dataset $D = \{(x_1, y_1), \ldots, (x_n, y_n)\}$, noise parameter $\lambda$, Bayesian post-processing
  mechanism $\text{BPP}(\mathbf{o}, \lambda, prior)$ defined in Eq. (3), optimizer $\text{Update}(model, input, soft\_labels)$

---

Initialize noised dataset $D^* \leftarrow \emptyset$
**for** $i \leftarrow 1$ **to** $n$ **do**   // noising labels
   |   $\mathbf{o}_i \leftarrow \text{OneHot}(y_i) + \text{Laplace}(\lambda)$
   |   $D^* \leftarrow D^* \cup \{(x_i, \mathbf{o}_i)\}$
**end**

Randomly initialize model $M$
**repeat**
   |   **for** $(x, \mathbf{o})$ **in** $D^*$ **do**
   |    |   $\mathbf{pred} \leftarrow M(x)$
   |    |   $\mathbf{post} \leftarrow \text{BPP}(\mathbf{o}, \lambda, \mathbf{pred})$
   |    |   $M \leftarrow \text{Update}(M, x, \mathbf{post})$
   |   **end**
**until** *convergence*;

---

**Output**: Model $M$

---

Following standard analysis (Dwork and Roth [14, Chapter 3.3]), $\varepsilon$-DP can be achieved with additive Laplace noise with per-coordinate standard deviation $2\sqrt{2}/\varepsilon$. Analogously, Gaussian noise sampled from $\mathcal{N}(0, \sigma^2 \cdot \mathbb{I}_C)$ results in $(\varepsilon, \delta)$-DP if $\sigma = \sqrt{2\ln(1.25/\delta)}/\varepsilon$ and $\varepsilon, \delta < 1$ ([14, Appendix A]).

### 5.2 Post-processing of Soft Randomized Response

The output of Soft-RR is no longer a valid soft label vector—values are not necessarily constrained to the [0,1] interval and do not sum up to 1. Thankfully, the fundamental property of DP allows privacy preservation under post-processing, which we use to obtain a probability mass function on perturbed labels to de-noise the mechanism's output. Treating the noisy labels $\mathbf{o}$ as the observables, and $\lambda$ as the noise parameter, we can calculate the posterior $p(y = c \mid \mathbf{o}, \lambda)$ by Bayes' rule as:

$$p(y = c \mid \mathbf{o}, \lambda) = \frac{p(\mathbf{o} \mid y = c, \lambda) \cdot p(y = c)}{\sum_k p(\mathbf{o} \mid y = k, \lambda) \cdot p(y = k)}. \tag{1}$$

In the context of SGD, the current model's output can be interpreted as a prior probability distribution over the classes $(p(y))$.

ALIBI, or Additive Laplace with Iterative Bayesian Inference, is an algorithm that applies Bayesian post-processing to the output of the Laplace mechanism (Algorithm 3). With Laplace noise, the output of the mechanism for a specific label $k$ is distributed as:

$$p(\mathbf{o} \mid y = k, \lambda) \propto e^{-|\mathbf{o}_k - 1|/\lambda} \prod_{j \neq k} e^{-|\mathbf{o}_j|/\lambda}. \tag{2}$$

Plugging (2) into (1) we derive:

$$p(y = c \mid \mathbf{o}, \lambda) = \text{SoftMax}(f(o_c)/\lambda + \log p(y = c)), \tag{3}$$

where $f(o_c) = -\sum_k |\mathbf{o}_k - [c = k]|$ and $[\cdot]$ is the Iverson bracket.

At a given privacy level, ALIBI empirically outperforms iterative Bayesian inference on additive Gaussian noise and naïve "uninformed" post-processing strategies, the details of which are deferred to Section D (Supplemental materials).

## 6 Evaluation

The algorithms are evaluated by training the Wide-ResNet architecture [38] on the CIFAR-10 and CIFAR-100 datasets [22]. The widening factor is set to 4 and 8 for CIFAR-10 and CIFAR-100 respectively. To facilitate comparison of privacy assured by the two approaches, we train our models

to achieve similar accuracy levels and provide the computed theoretical $(\varepsilon, \delta)$ upper bound. (Ignoring the privacy costs of hyperparameter tuning and of reporting the test set performance. The former can be minimized, with a modest loss in accuracy, by applying the selection procedure of Liu and Talwar [24].) We also provide the empirical privacy loss lower bound, $\varepsilon_m$, as estimated by the black-box attacks (Section 7).

PyTorch-based implementations of ALIBI, PATE-FM, and memorization attacks, including configuration options and hyperparameters, are publicly available [1].

## 6.1 PATE-FM

PATE-FM parameters are given in Table 3 (Supplemental materials). The student and the teacher models share the same architecture and are trained using FixMatch [32, 21] with hyperparameters from Sohn et al. [32] (we use the RandAugment variant [9]). The number of epochs is set so that the test accuracy reaches a plateau: for CIFAR-10, we train teachers for 40 epochs and the student for 200 epochs; for CIFAR-100, we train both the teachers and the student for 125 epochs. To report privacy, we set $\delta = 10^{-5}$. The student's accuracy is reported as the average of three runs (for a fixed teacher ensemble).

There are several hyperparameters affecting privacy/utility trade-off in PATE: number of teachers, level of noise, voting threshold, and number of labeled samples in the student dataset. Apart from the expected dependencies (increasing noise and the number of teachers benefits privacy at some accuracy cost), we observe the following:

- The number of labeled student samples beyond a certain level (10–25 samples per class) has little effect on the eventual model accuracy. This is in contrast with the standard (non-private) FixMatch where increasing the number of labeled samples usually improves accuracy.
- The voting threshold $\tau$ needs to be kept between 10% and 75% of the overall number of teachers. As pointed out in Papernot et al. [29], a lack of consensus between teachers hurts both privacy and utility, as the teacher ensemble errs more often on such inputs and a single vote can have more impact on the outcome. Setting the threshold too high leads to a selection bias in training samples, forcing the student model to train only on the most trivial samples. Even in the zero-noise regime, the threshold above 75% considerably hurts the accuracy.

## 6.2 ALIBI

We use mini-batch SGD with a batch size of 128 and a momentum of 0.9. We train all models for 200 epochs and pick the best checkpoint. We tune the learning rate, weight decay and noise multiplier to obtain the desired accuracy while minimizing privacy $\varepsilon$ (see Table 4 in Supplemental materials). Since ALIBI is based on the Laplace mechanism, it achieves pure DP corresponding to $\delta = 0$.

## 6.3 Results

Table 1 presents the privacy budget and the memorization metric at three levels of accuracy across our two approaches for the both datasets. Additionally, for ALIBI, we provide the results of a higher accuracy model on the CIFAR-100 task; this higher accuracy was not attainable with PATE-FM.

For correct comparison with LP-2ST on CIFAR-100, we factor out modelling differences by applying both mechanisms to training the same architecture—ResNet18 [19, 25] with appropriate modifications to suit the task. Results are presented in Table 2. For reference, non-private baselines with this model achieve $\approx 95.5\%$ on CIFAR-10 and $\approx 79\%$ on CIFAR-100.

We note that there is a significant difference in computational resources required to train with PATE-FM and ALIBI. The reasons are twofold. First, in our experiments, models trained in a semi-supervised regime with few labeled examples take longer to converge. Second, PATE requires training the teacher ensemble, in which computational costs scale linearly with the size of the ensemble. Overall, this accounts for PATE-FM consuming 1,000–1,500$\times$ more computational resources (GPU·hr) compared to ALIBI.

Table 1: Privacy of PATE-FM [28, 32] and ALIBI, on CIFAR-10 and CIFAR-100 using Wide-ResNet18, matched by test accuracy levels. $\varepsilon$ for PATE is at $\delta = 10^{-5}$. Empirical privacy loss $\varepsilon_m$ is reported as a 95% confidence interval (CI).

| Dataset | Accuracy level | Algorithm | Accuracy | $\varepsilon$ | 95%-CI $\varepsilon_m$ |
|---|---|---|---|---|---|
| CIFAR-10 | High | PATE-FM | 93.7% | 1.6 | 0.0–0.9 |
| | | ALIBI | 94.0% | 8.0 | 2.9–4.0 |
| | Medium | PATE-FM | 86.9% | 0.29 | 0.0–0.6 |
| | | ALIBI | 84.2% | 2.1 | 1.0–2.2 |
| | Low | PATE-FM | 73.4% | 0.18 | 0.0–0.4 |
| | | ALIBI | 71.0% | 1.0 | 0.0–2.2 |
| CIFAR-100 | Very High | PATE-FM | – | – | – |
| | | ALIBI | 75.3% | 8.1 | 3.5–4.5 |
| | High | PATE-FM | 69.9% | 715 | 0.4–1.0 |
| | | ALIBI | 71.4% | 6.3 | 2.8–3.5 |
| | Medium | PATE-FM | 50.0% | 16 | 0.2–0.7 |
| | | ALIBI | 51.6% | 3.0 | 1.4–2.4 |
| | Low | PATE-FM | 30.5% | 7.9 | 0.2–0.6 |
| | | ALIBI | 31.4% | 2.0 | 0.6–1.0 |

Table 2: Test accuracy of ALIBI and LP-2ST [18, Table 4], on CIFAR-100 applied to ResNet18, matched by $\varepsilon$.

| Algorithm | $\varepsilon = 3$ | $\varepsilon = 4$ | $\varepsilon = 5$ | $\varepsilon = 6$ | $\varepsilon = 8$ |
|---|---|---|---|---|---|
| ALIBI | **55.0** | **65.0** | **69.0** | **71.5** | **74.4** |
| LP-2ST | 28.7 | 50.2 | 63.5 | 70.6 | 74.1 |

# 7 Measuring Memorization

We complement our evaluation with an empirical study of the (label) memorization abilities of classifiers trained with PATE-FM and ALIBI. While we can derive *provable* upper-bounds on the level of $\varepsilon$-Label DP provided by both algorithms, a direct comparison between the (actual) privacy of both algorithms is challenging as they rely on very different privacy analyses. As we will see, for CIFAR-100, PATE-FM empirically appears to provide much stronger privacy guarantees than ALIBI, even though the privacy analysis suggests the opposite.

To empirically measure privacy loss, we effectively instantiate the implicit adversary in the DP definition [26, 20]. We construct two datasets $D, D'$ that differ only in the label of one example, and train a model on one of these two datasets. We then build an attack that "guesses" which of the two datasets was used. The attacker's advantage over a random guess can be used to derive an empirical lower bound on the mechanism's differential privacy. A formal treatment of memorization attacks using the framework of security games appears in Section B (Supplemental materials).

We point out the following differences between our approach and prior work. When providing lower bounds on privacy leakage, it is appropriate to consider the most powerful attacker pursuing the least challenging goal. In the context of standard DP (privacy for labels and features), it means an attacker that can add or remove a sample and seeks to determine whether the sample was part of the training dataset (i.e., membership inference [31]). In contrast, we operate within the constraints of Label DP, where the attacker's power is limited to manipulating a single label and the goal is to infer which label was used during training. (The closest analogue is attribute inference [37].)

Prior work replicates a memorization experiment many times (e.g., 1,000 times [26]) to lower bound $\varepsilon$ with standard statistical tools [26, 20]. As this approach is prohibitively expensive in our setting (prior work computed DP lower bounds for simpler datasets such as MNIST), we introduce the following heuristic: we train a *single* model on a training set with 1,000 labels randomly flipped to a different class. For each of these "canaries" [6] with (flipped) label $y'$, the attacker has to guess whether

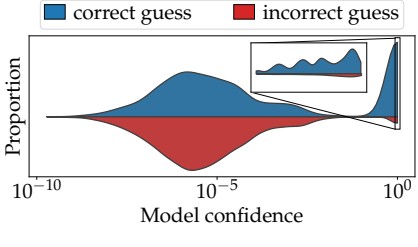

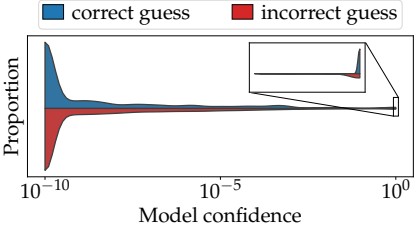

| (a) ALIBI (CIFAR-100, High) | (b) PATE-FM (CIFAR-100, High) |

Figure 1: Comparison of the adversary's success rates in guessing the label of inserted canaries with ALIBI and PATE-FM (for CIFAR-100, with High accuracy level). We sort canaries by the model's maximal confidence on the two labels $y', y''$ that the adversary has to guess from. The violin plots show the proportion of correct guesses (the model is more confident in the canary label) and incorrect guesses. To maximize the adversary's distinguishing power, we limit the adversary to only issuing guesses for canaries where the model's confidence in either $y'$ or $y''$ is above $99\%$ (zoomed-in box). ALIBI is clearly more vulnerable to the attack (i.e., it is easier for the adversary to guess a canary's label), even though its theoretical privacy analysis yields a much lower bound on $\varepsilon$ than for PATE-FM.

the canary's assigned label is $y'$ or $y''$, where $y'' \neq y'$ is a different random incorrect label. If the adversary has accuracy $\alpha$ in this guessing game, we can lower bound $\varepsilon$ by $1 - \log(\frac{\alpha}{1-\alpha})$. As in prior work [26, 20], we compute a $95\%$ confidence interval on the adversary's guessing accuracy $\alpha$ and thus on the level of privacy $\varepsilon$ against our attack. We formally define our attacking game and analyze the relationship between the attacker's success and the privacy budget $\varepsilon$ in Section B (Supplemental materials).

To isolate the impact that can be attributed to the difference between privacy-preserving training procedures, we apply ALIBI and PATE-FM to the same target model architecture and tune their privacy parameters to achieve similar accuracy levels. (For discussion of confounders, see Erlingsson et al. [15].) Furthermore, we deliberately limit memorization attacks to be trainer-agnostic, to avoid privileging, for instance, a more complex training procedure over a simpler one.

Our empirical estimates of the privacy loss $\varepsilon_m$ are given in Table 1. The estimates for ALIBI are approximately within a factor of 2 of the theoretical upper bound on $\varepsilon$. This suggests that the theoretical privacy analysis of ALIBI is close to tight. On the other hand, for PATE-FM our empirical estimates of the privacy loss are up to three orders of magnitude smaller than the theoretical upper bound. While we have no guarantee that our attack is the strongest possible (and it likely is not), our experiment does suggest that for a fixed accuracy level, PATE-FM is much less likely to memorize labels compared to ALIBI (see Figure 1). Empirically, it thus appears that PATE-FM provides a much better privacy-utility tradeoff than ALIBI, even though the theoretical analysis suggests the opposite.

This being said, we encourage critical interpretation of privacy claims backed by memorization estimates. Attacks, and the corresponding lower bounds on privacy loss, are provisional and conditional; they can be shattered with better algorithms or additional resources. The empirical lower bounds should be contrasted with provable upper bounds that provide the most conservative guidance in selecting privacy-preserving algorithms and setting their parameters.

## 8    Conclusion

We have proposed and evaluated two approaches toward achieving Label DP in ML: ALIBI and PATE-FM (the PATE framework instantiated with FixMatch). We have demonstrated that both approaches can achieve state-of-the-art results on standard datasets. As the two approaches have different theoretical foundations and exhibit very different performance characteristics, they also present an interesting combination of trade-offs.

PATE-FM offers impressive empirical privacy despite a very conservative theoretical privacy analysis. On the other hand, its training setup is fairly complicated. The dependency on semi-supervised learning techniques further restricts the tasks this method can be used for at present.

In contrast, ALIBI's implementation and interpretation are quite simple. (1) It is a generic algorithm that is compatible with any iterative procedure for ML training, such as SGD or its variants. It is not a

black-box however, and requires the ability to update training data labels as they are processed by the training algorithm. (2) It enjoys a tighter privacy analysis resulting in a more realistic upper bound on the privacy budget, but appears to offer less privacy than PATE-FM empirically. (3) It can be used to train models to accuracy levels which PATE cannot achieve without significantly more data.

In addition to privacy and performance trade-offs, our two algorithms also differ significantly in their time and computational resource requirements (as discussed in the Section 6.3).

Beyond the concrete improvements offered by our new techniques, our work emphasizes the need for more research across multiple avenues: (1) bridging the gap between privacy lower bounds, backed by attacks, and upper bounds, based on privacy analyses (in particular for PATE); (2) making the privacy analysis of different approaches comparable; (3) designing stronger attack models to better capture empirical privacy guarantees.

## 9    Acknowledgements and Funding Sources

The authors are grateful to NeurIPS anonymous reviewers and area chairs for their helpful comments. Florian Tramèr is supported by NSF award CNS-1804222.

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
