## A Hyperparameters

Tables 3 and 4 summarize hyperparameters for PATE-FM and ALIBI respectively.

Table 3: PATE-FM (Algorithms 1 and 2) hyperparameters for select accuracy levels.

| Dataset | Teachers | $\sigma_1$ | $\sigma_2$ | $\tau$ | Queries answered | Accuracy Ensemble | Student |
|---|---|---|---|---|---|---|---|
| | 200 | 160 | 20 | 100 | 500 | 90.8% | 93.7% |
| CIFAR-10 | 800 | 800 | 300 | 400 | 250 | 60.8% | 86.9% |
| | 800 | 800 | 500 | 400 | 250 | 36.4% | 73.4% |
| | 20 | 7 | 2 | 2 | 9,400 | 74.0% | 69.9% |
| CIFAR-100 | 100 | 45 | 15 | 10 | 1,000 | 55.0% | 50.0% |
| | 100 | 90 | 30 | 10 | 1,000 | 28.8% | 30.5% |

Table 4: Hyperparameters of ALIBI (Algorithm 3) for select accuracy levels.

| Dataset | Learning rate | Weight decay $(\times 10^{-4})$ | Accuracy |
|---|---|---|---|
| | 0.308 | 0.568 | 94.0% |
| CIFAR-10 | 0.960 | 0.00796 | 84.2% |
| | 0.315 | 1.50 | 71.0% |
| | 0.0037 | 33.5 | 75.3% |
| CIFAR-100 | 0.0057 | 17.5 | 71.4% |
| | 0.100 | 2.33 | 51.6% |
| | 0.1 | 1.00 | 31.4% |

## B Memorization: Formal treatment

To empirically bound the level $\varepsilon$ of DP, prior work instantiates a general *membership inference* game, defined in Figure 2 for two arbitrary neighboring datasets $D_0$ and $D_1$.

Figure 2: Basic membership inference game, Game 1.

By repeating this game multiple times, we can estimate the adversary's success rate and convert this into a lower bound on $\varepsilon$.

This would be prohibitively expensive in our setting (each iteration of the game requires training a model on CIFAR-10 or CIFAR-100, and the game has to be repeated about 1,000 times to get

non-trivial bounds). We thus propose a heuristic approach for running multiple iterations of this game while training a *single* model.

First, we will change the game slightly so as to allow the adversary to *abstain* from issuing a guess on some instances. That is, the output range of $\mathcal{A}$ is $\{0, 1, \perp\}$. We then define the adversary's *correct guess rate* (CGR) as:

$$\text{CGR}_{D_0,D_1} := \Pr[b = b' \mid b' = \mathcal{A}(D_0, D_1, \mathcal{M}(D_b)) \wedge b' \neq \perp] .$$

The probability is taken over the bit $b$, the randomness of the mechanism $\mathcal{M}$ and the algorithm $\mathcal{A}$.

**Theorem B.1.** *If $\mathcal{M}$ satisfies $\varepsilon$-DP and $D_0, D_1$ are two adjacent databases, then*

$$\varepsilon \geq \log\left(\frac{\text{CGR}_{D_0,D_1}}{1 - \text{CGR}_{D_0,D_1}}\right) .$$

*Proof.* For notational convenience, define $A_x$ as a random variable distributed according to $\mathcal{A}(D_0, D_1, \mathcal{M}(D_x))$ for $x \in \{0, 1, b\}$. Then

$$
\begin{aligned}
\frac{\text{CGR}_{D_0,D_1}}{1 - \text{CGR}_{D_0,D_1}} &= \frac{\Pr[b' = b \mid b' = A_b \wedge b' \neq \perp]}{\Pr[b' = 1 - b \mid b' = A_b \wedge b' \neq \perp]} \\
&= \frac{\Pr[A_b = b]}{\Pr[A_b = 1 - b]} \\
&= \frac{\Pr[A_0 = 0] + \Pr[A_1 = 1]}{\Pr[A_0 = 1] + \Pr[A_1 = 0]} \\
&\leq \max\left\{\frac{\Pr[A_0 = 0]}{\Pr[A_1 = 0]}, \frac{\Pr[A_0 = 1]}{\Pr[A_1 = 1]}\right\} \qquad \text{(by the mediant inequality)} \\
&\leq e^\varepsilon ,
\end{aligned}
$$

where the last inequality follows from the assumption that $\mathcal{M}$ is $\varepsilon$-DP. $\qquad\square$

It now remains to be seen how we can bound the adversary's correct guessing rate CGR. We define Game 2 (Figure 3) by making a small change to Game 1 above, so that the neighboring datasets $D_0$ and $D_1$ are chosen *at random* in each iteration of the game, by flipping the label of one example of a common dataset $D$.

| Attacker | Challenger($D$) |
|---|---|
| | $D_0, D_1 \leftarrow D$ |
| | $i \leftarrow^{\$} [1 : \lvert D \rvert]$ |
| | $y', y'' \leftarrow^{\$} \{[1 : C] \setminus \{y_i\}\}$, s.t. $y' \neq y''$ |
| | $(D_0)_i \leftarrow (x_i, y')$ |
| | $(D_1)_i \leftarrow (x_i, y'')$ |
| | $b \leftarrow^{\$} \{0, 1\}$ |
| | $M \leftarrow \mathcal{M}(D_b)$ |
| | $\xleftarrow{\quad D_0, D_1, M \quad}$ |
| $b' \leftarrow \mathcal{A}(D_0, D_1, M)$ | |
| | $\xrightarrow{\quad b' \quad}$ |
| | output $b = b'$ |

Figure 3: $D_0$ and $D_1$ are defined randomly (Game 2).

Similar to Game 1, we define the adversary's probability of winning in Game 2 conditional on $\mathcal{A}$'s output not being $\perp$. Let the *average* CGR in Game 2 (ACGR) be:

$$\mathrm{ACGR}_D := \mathop{\mathbb{E}}_{D_0,D_1} \left[ \Pr[b = b' \mid b' = \mathcal{A}(D_0, D_1, \mathcal{M}(D_b)) \wedge b' \neq \perp] \right]$$

$$= \mathop{\mathbb{E}}_{D_0,D_1} \left[ \mathrm{CGR}_{D_0,D_1} \right] .$$

If we can lower-bound ACGR by some value $\alpha$, then there exists at least one pair of neighboring datasets $D_0$, $D_1$ such that $\mathrm{CGR}_{D_0,D1} \geq \alpha$. As the DP guarantee has to hold for *all* neighboring datasets, we can use a bound on ACGR to bound $\varepsilon$.

Finally, instead of repeating Game 2 many times to get a bound on ACGR, we instead simulate multiple iterations of Game 2 at once, which becomes Game 3 (formally defined in Figure 4). The

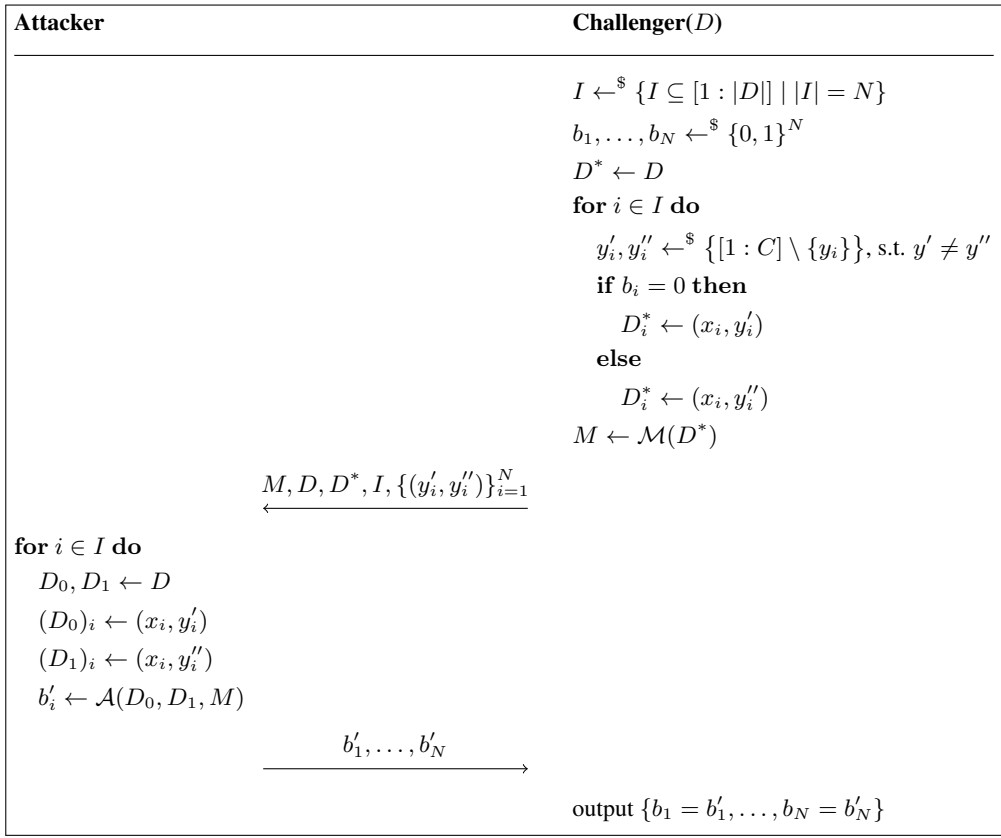

Figure 4: Game 3.

heuristic step here is that we assume that each of the $N$ guesses made by the adversary are independent from each other and reflect the adversary's guesses in $N$ independent iterations of Game 2.

Given one iteration of Game 3 with $N$ "canaries", we can compute a lower bound on the adversary's average CGR using standard confidence intervals for a Binomial random variable. That is, we count the number of correct guesses among the $M \leq N$ instances where the adversary made a guess $b_i' \neq \perp$, and apply a Clopper-Pearson bound.

We can improve the tightness of this bound further. In Game 3, for each canary $(x_i, y_i')$ that the model is trained on (assuming $b_i = 0$), we record the adversary's guess with respect to only one other random label $y''$. Yet, we could record the adversary's guess with respect to *all* $C - 2$ choices of $y_i'' \neq y_i, y_i'$ to get a tighter estimate of the adversary's average success rate. However, we definitely cannot treat these guesses as independent. Instead, we first estimate the adversary's (empirical) average correct guessing rate $\overline{\mathrm{ACGR}}_i$ for each canary (where the average is taken over all possible

choices for $y_i''$):

$$b_{i,j}' := \mathcal{A}(D_{b_{i,j}}^{(i,j)}, D_{1-b_{i,j}}^{(i,j)}, M) \quad \text{for } j = 1, \dots, C-2,$$

$$m_i := \sum_{j=1}^{C-2} [b_{i,j}' \neq \perp],$$

$$\overline{\text{ACGR}}_i := \frac{1}{m_i} \sum_{j=1}^{C-2} [b_{i,j}' = b_{i,j}],$$

where $b_{i,j}$ are iid uniform bits, $D_0^{(i,j)}$ are $D$ for all $i$ and $j$, $D_1^{(i,j)}, \dots, D_1^{(i,j)}$ are copies of $D$ with the $C-2$ possible choices for the label $y_i''$, and $[\cdot]$ is the Iverson bracket. (By convention, $0/0 = 0$.) We then compute a confidence interval for the empirical mean of all the $\overline{\text{ACGR}}_i$. As the adversary may abstain from making a guess with a different probability for each canary (i.e., the $m_i$'s may not all be equal) we have to weigh the $\overline{\text{ACGR}}_i$ values accordingly. That is, we compute the *weighted* average and standard deviation of the $\overline{\text{ACGR}}_i$ with the $M_i$ as *reliability weights*. Finally, we obtain a confidence interval for the adversary's $\overline{\text{ACGR}}$ using a standard 95% confidence interval for the normal distribution.

Finally, it remains to define our adversary $\mathcal{A}(D_0, D_1, M)$ where $(x_i, y') \in D_0$ and $(x_i, y'') \in D_1$. The adversary simply looks at the model's confidence on $x_i$ for both possible labels and guesses that the more confident of the two is the label that the model was trained on. However, if both labels have confidence below some fixed threshold $\tau$, the adversary abstains. Formally:

$$\mathcal{A}(D_0, D_1, M) = \begin{cases} \perp & \text{if } \max(M(x_i)_{y'}, M(x_i)_{y''}) < \tau \\ [M(x_i)_{y'} > M(x_i)_{y''}] & \text{otherwise} \end{cases}.$$

We consider different thresholds $\tau \in [0.5, 0.99]$ and report the best resulting attack (i.e., the setting with the highest lower-bound on $\overline{\text{ACGR}}$). For simplicity, we omit corrections for multiple hypothesis testing.

## C   Memorization: Validating the Heuristic

The previous section introduces a sequence of security games (Figures 2–4) that relate the success probability of a membership inference adversary (Game 1) to an efficient computational procedure (Game 3). It starts by randomizing a single instance of Game 1 into Game 2. In order to compute the adversary's probability of winning Game 2 with sufficient accuracy, the experiment needs to be repeated hundreds of times. Doing so would be prohibitively expensive as each run requires training a new model from scratch.

We use a heuristic whereby the independent runs of Game 2 are replaced with correlated instances of the membership inference game that share the same trained model (Game 3). Concretely, it means that instead of introducing a single canary (a mislabeled input) into a training dataset, Game 3 injects multiple canaries all at once. While doing so does change the input distribution of the training procedure, we argue that the overall effects are minimal and do not qualitatively affect our findings.

We test the heuristic's validity by comparing the adversary's advantage against ALIBI in Game 3, where the number of simultaneously inserted canaries is $N = 1,000$ as in Table 1, with ten repetitions of Game 3 with $N = 100$. The results are presented in Table 5. Notably, the 95% confidence intervals for $\varepsilon_m$ are in very close agreement, thus supporting the heuristic.

## D   Post-processing for Soft Randomized Response

For completeness, we describe Soft RR variants instantiated with uninformed post-processing and the Gaussian mechanism. We found that ALIBI dominates alternatives by achieving better accuracy with stronger privacy. In particular, ALIBI's upper bounds are stronger than those of the Gaussian mechanisms for the same levels of accuracy, while their empirical privacy losses are statistically indistinguishable.

Table 5: Comparison of 95% confidence intervals (CI) for the membership inference adversary against ALIBI given a single run of Game 3 with $N = 1000$ and 10 runs of Game 3 with $N = 100$.

| Dataset | Accuracy level | 95%-CI $\varepsilon_m$ | |
| | | 1,000 labels | 10×100 labels |
| --- | --- | --- | --- |
| | High | 2.9–4.0 | 2.7–3.6 |
| CIFAR-10 | Medium | 1.0–2.2 | 0.2–2.5 |
| | Low | 0.0–2.2 | 0.0–1.6 |
| | High | 2.8–3.5 | 2.7–3.4 |
| CIFAR-100 | Medium | 1.4–2.4 | 1.3–2.0 |
| | Low | 0.6–1.0 | 0.6–1.1 |

### D.1 Uninformed post-processing

Given Soft-RR's output vector $\mathbf{o}$, we may reduce error by mapping it to the closest point on the probability simplex (compare with Nikolov et al. [27]). In other words, we are solving the following constrained optimization problem:

$$\min \|\mathbf{o}^* - \mathbf{o}\|_2 \quad \text{subject to} \quad \begin{cases} \forall i \;\; 0 \le o_i^* \le 1, \\ \sum_i o_i^* = 1 \end{cases}$$

The problem is (strictly) convex, and thus admits a unique, efficiently computable solution. Moreover, a particularly simple and efficient method (Algorithm 4) exists due to Duchi et al. [10] (see also Wang and Carreira-Perpiñán for other approaches [35]).

---

**Algorithm 4:** Post-processing using Min Projection (Duchi et al. [10]).

**Input**: $\mathbf{o} = (o_1, \ldots, o_C) \in \mathbb{R}^C$
**Output**: Projection of $\mathbf{o}$ onto the probability simplex
Sort $\mathbf{o}$ as $s_1 \ge s_2 \ge \cdots \ge s_C$
Find $k \leftarrow \max_j \left\{ j \in [1:C] \colon s_j > \frac{1}{j}\left(\sum_{i=1}^{j} s_i - 1\right) \right\}$
$u \leftarrow \frac{1}{k}\left(\sum_{i=1}^{k} s_i - 1\right)$
**for** $i \leftarrow 1$ **to** $C$ **do**
$\quad | \quad o_i' \leftarrow \max(o_i - u, 0)$
**end**
Output $\mathbf{o}'$

---

### D.2 Bayesian post-processing on Additive Gaussian Mechanism

This follows the same post-processing algorithm as ALIBI as described in Section 5.2, except for Gaussian noise instead of Laplace. Eq. (2) changes to the following (note the switch of the noise parameter from $\lambda$ to $\sigma$):

$$p(\mathbf{o} \mid y = k, \sigma) \propto e^{-\frac{(\mathbf{o}_k - 1)^2}{2\sigma^2}} \prod_{j \neq k} e^{-\frac{\mathbf{o}_j^2}{2\sigma^2}}. \tag{4}$$

Plugging (4) in (1) we have:

$$p(y = c \mid \mathbf{o}, \sigma) = \frac{e^{\mathbf{o}_c/\sigma^2} \cdot p(y = c)}{\sum_k e^{\mathbf{o}_k/\sigma^2} \cdot p(y = k)} = \text{SoftMax}(\mathbf{o}_c/\sigma^2 + \log p(y = c)). \tag{5}$$

The training algorithm as described in Algorithm 3 can be modified by setting BPP to (5) to work in this setting.

The results of applying the Gaussian mechanism ("AGIBI") are reported in Table 6 together with ALIBI performance from Table 1 for ease of comparison. The claimed privacy losses (the $\varepsilon$ column) strongly favor ALIBI over the Gaussian mechanism; the empirically computed privacy loss lower bounds do not separate the two mechanisms.

Table 6: ALIBI and AGIBI on CIFAR-10 and CIFAR-100 using Wide-ResNet18, matched by test accuracy levels. Empirical privacy loss $\varepsilon_m$ is reported as a 95% confidence interval (CI).

| Dataset | Accuracy level | Algorithm | Accuracy | $\varepsilon$ | 95%-CI $\varepsilon_m$ |
|---|---|---|---|---|---|
| CIFAR-10 | High | ALIBI | 94.0% | 8.0 | 2.9–4.0 |
| | | AGIBI | 93.5% | 19 | 2.1–3.2 |
| | Medium | ALIBI | 84.2% | 2.1 | 1.0–2.2 |
| | | AGIBI | 84.3% | 7.7 | 0.7–1.4 |
| | Low | ALIBI | 71.0% | 1.0 | 0.0–2.2 |
| | | AGIBI | 71.3% | 3.7 | 0.0–2.0 |
| CIFAR-100 | High | ALIBI | 71.4% | 6.3 | 2.8–3.5 |
| | | AGIBI | 69.9% | 17 | 2.7–3.4 |
| | Medium | ALIBI | 51.6% | 3.0 | 1.4–2.4 |
| | | AGIBI | 50.8% | 8.4 | 1.3–2.0 |
| | Low | ALIBI | 31.4% | 2.0 | 0.6–1.0 |
| | | AGIBI | 28.7% | 5.4 | 0.6–1.1 |