# OpenReview forum: "Antipodes of Label Differential Privacy: PATE and ALIBI"
_NeurIPS.cc/2021/Conference — NeurIPS 2021 Poster_

### Official Review · Reviewer_zWa1 · 2021-07-11

**Rating:** 6
**Confidence:** 4

**Summary:**

This paper focuses on deep learning with level differential privacy (DP), in which the privacy of only the label is preserved. The authors propose two new algorithms: a variation of PATE using a recent semi-supervised learning approach and the fact that features are public in label DP to train better teacher models and use fewer student model queries which consume DP budget ; and a new approach based on local DP noise for the labels, and a Bayesian post-processing that uses the current version of the model to refine guesses about the true underlying label. Finally, the authors run a label memorization attack to empirically assess the privacy.


**Main Review:**

The paper is interesting, and proposes neat algorithmic improvements that seem like they would be useful. The fact that ALIBI supports a local DP type interaction is also promising. However there are two points that would gain to being studied in more depth.

The point that different DP analysis tightness can lead to the choice of less private models feels both obvious and somewhat irrelevant. The whole point of DP is to provide an upper-bound on the privacy loss: of course if that bound is not tight, another algorithm with a tighter bound could look better. However, empirical measures of privacy like the one discussed in the paper, which is a lower bound on the privacy loss, does not seem like a good metric to decide which algorithm to chose. The discussion should emphasize that more, even earlier in the paper.
The empirical observation seems interesting, in particular with the switch between CIFAR-10 and 100, but it should be studied in more depth. Is there a reason to expect the analysis to break down in that case, maybe dure to the larger number of classes? The discussion would be more interesting if you could pinpoint the slackness, and see if there might be hopes to fix it. This would make the gap much more interesting.

It would be interesting to analyze the ALIBI algorithm a bit more formally. For instance, can it be seen as some type of EM or empirical Bayes procedure, and what would be the implications of that?

Evaluation:
- For ALIBI, how do you choose the best checkpoint with DP, and how is it accounted for in the total budget?
- It would be important to have some details about the attacks used for the empirical DP evaluation.


**Time Spent Reviewing:**

2.5h

---

> ### Author Response · Authors · 2021-08-10
> **Response to review**
>
> > However, empirical measures of privacy like the one discussed in the paper, which is a lower bound on the privacy loss, does not seem like a good metric to decide which algorithm to choose. The discussion should emphasize that more, even earlier in the paper.
>
> Our motivation behind and intention of proposing empirical analysis via attacks were two-fold:
> 1. To enable the comparison of privacy techniques whose epsilon values are produced by different analyses and thus may not be directly comparable.
> 2. To indicate that a high theoretical upper bound does not necessarily imply the mechanism is less private. On the contrary, it invites us to tighten the analysis. In other words, while a lower bound may not enable us to choose one algorithm over the other, it will hopefully discourage us from dismissing one. Examples of this are the CIFAR-100’s  low- and medium-accuracy regimes, where PATE-FM seemed to have memorized less than ALIBI despite a significantly higher epsilon.
>
> We will affirm the importance of provable bounds throughout the full version of the paper.
>
> > The empirical observation seems interesting, in particular with the switch between CIFAR-10 and 100, but it should be studied in more depth. Is there a reason to expect the analysis to break down in that case, maybe due to the larger number of classes?
>
> The answer is elaborated in our response to [Reviewer 1](https://openreview.net/forum?id=sR1XB9-F-rv&noteId=gvWnQ6RPDS4). Indeed, PATE’s performance degrades rapidly with the number of classes due to the combined impact of having fewer teachers and requiring more queries to train the student.
>
> > For ALIBI, how do you choose the best checkpoint with DP, and how is it accounted for  in the total budget?
>
> We presume “checkpoint” here refers to the classical ML definition of a saved model state. If this presumption is true, we pick the checkpoint that yields the best accuracy on the test set. Note that, unlike DP-SGD, the privacy cost of the training algorithm is fixed at the beginning of the training (once the labels are perturbed with the additive noise) and it does not grow with time.
>
> The only dependency on the number of epochs is in the privacy budget of the evaluation step. Although we do not explicitly compute it, it tends to be minimal as the accuracy is computed over 10,000 images. (Liu and Talwar, https://arxiv.org/abs/1811.07971, discuss how this dependency can be minimized at a small utility loss.)
>
> If, however, “checkpoint” here refers to the checkpoint as defined in Ghazi et al, it doesn’t apply to our work. In fact, we distinguish ourselves from the work of Ghazi et al. in not requiring dataset partitioning. We de-noise the labels by post-processing, which integrates seamlessly with the conventional training process.

---

> > ### Author Response · Authors · 2021-08-19
> > **Have concerns been addressed?**
> >
> > We'd like to follow up on our response, and especially on our interpretation of the word "checkpoint". Does it adequately answer the reviewer's question?

---

> > > ### Comment · Reviewer_zWa1 · 2021-08-19
> > > **Reply**
> > >
> > > Thank you for your answer. I did refer to the first checkpoint definition, and was indeed worried about the privacy budget of the evaluation step. It would be nice to explicitly state that it is not accounted for in the paper to avoid confusion, and maybe give an example of its cost (even under the most basic approach) in one experiment.

---

> > > > ### Author Response · Authors · 2021-08-20
> > > > **Privacy budgeting for evaluation step**
> > > >
> > > > We will definitely state that our privacy budget does not include reporting of accuracy numbers (keeping with the standard practice in the literature). To consider a concrete example, computing accuracy with standard deviation of 0.1% can be done with eps=0.1 (via the Laplace mechanism applied to a sum over 10,000 bits). Reporting accuracy 25 times (every 5 epochs until 125'th epoch) would have the total budget of eps = 2.2 at delta = 10^-5 via the advanced composition theorem.

---

### Official Review · Reviewer_bt7g · 2021-07-11

**Rating:** 7
**Confidence:** 4

**Summary:**

The authors consider label differential privacy (DP) where the goal is to satisfy DP with
respect to the labels of the examples. This model was formally introduced by Chaudhuri
and Hsu (COLT 2011).

They propose two models for label DP:

1) Based on ALIBI (Additive Laplace noIse coupled with Bayesian Inference)

2) Based on PATE (Private Aggregation of Teacher Ensembles)

A prominent example where label DP is quite useful is in online advertising where the goal is to,
given a user's profile, predict the advert conversion probability (the label). If we assume that
the training is done by the advertising network (e.g., a tech company) who has the data (the features),
the goal is to make sure the advertiser (some company who wishes to advertise on a tech platform)
cannot infer the labels.

They estimate the *empirical* privacy loss of the trained models via the lens of black-box
membership inference attacks.

The novelty of this work, it seems, lies in the (previously unexplored) connections between
PATE and label DP. Ghazi et al. also proposed randomized response mechanisms for label privacy.

They experimentally evaluate the performance of their algorithms on CIFAR-10, CIFAR-100 where
they show medium to high accuracy results
(e.g., 93.7% on $\epsilon = 1.6$).



**Ethics Review Area:**

["I don’t know"]

**Limitations And Societal Impact:**

Yes, they adequately discuss limitations of their work. See their conclusions where they state that their techniques could lead to more work that can help with "bridging the gap between privacy lower bounds, backed by attacks, and upper bounds, based on privacy analyses (in particular for PATE); (2) making the privacy analysis of different approaches comparable; (3) designing stronger attack models to better communicate empirical privacy guarantees."

**Main Review:**

The authors propose label privacy mechanisms based on ALIBI (and soft randomized response)
and PATE.

The PATE Framework
------------------

PATE splits learning into stages. First, a number of "teacher"s are trained on disjoint
partitions of the non-public labeled dataset. Second, a "student" trains its own model by querying the
"teacher"s on samples drawn from public unlabeled data. DP guarantees are enforced at the
communication points between the teachers and the student.
Because of the disjointness of the "teacher"'s data, one private sample affects at most one
"teacher".

Their Adaptation of PATE:
Each teacher gets $n/T$ labeled examples and $n$ unlabeled examples. And $K$ is the number of labels
requested by the student. Note that the choice of $T, K$ results in two competing objectives.

Randomized Response and ALIBI
-----------------------------

Ghazi et al. already proposed a randomized response mechanism. In this framework, the true label is
predicted with probability $p$ and the other label (assuming we're in the binary case)
is predicted with probability $1-p$. The authors deviate from previous work in the use of
one-hot encoding of the training sample label (Soft-RR).
Because the output of Soft-RR results in values not necessarily in [0, 1] they use
post-processing and Bayesian inference.



Minor Comments
--------------
- In Table 1, 2 the accuracy of ALIBI and PATE-FM are shown. Can you also include the non-private
accuracies? [response acknowledged]

- $\epsilon = 715?$ in the PATE-FM high accuracy level in table 1? [response acknowledged]


**Time Spent Reviewing:**

4 hours

---

> ### Author Response · Authors · 2021-08-10
> **Response to review**
>
> > In Table 1, 2 the accuracy of ALIBI and PATE-FM are shown. Can you also include the non-private accuracies?
>
> Non-private WideResNet18 achieves ~95.5% on CIFAR-10 and ~79% on CIFAR-100. For comparison, SotA of training with full (features-and-labels) privacy is 74.9% accuracy on CIFAR-10 at $\epsilon=8$, $\delta=10^{−5}$ (https://arxiv.org/pdf/2102.12677.pdf).
>
> > eps = 715? in the PATE-FM high accuracy level in table 1?
>
> This is not a typo. The best known privacy analysis of PATE-FM does yield $\epsilon = 715$. Improving PATE privacy claims is beyond the scope of the present paper, although it is not hard to point out where the current analysis may be overly pessimistic. For instance, PATE assumes that changing the label of a _single_ example may flip _all_ votes of the teacher tainted by that example in the _worst possible_ manner.

---

### Official Review · Reviewer_ufMh · 2021-07-16

**Rating:** 6
**Confidence:** 4

**Summary:**

 The paper considers the label DP setting and proposes ALIBI and PATE-FM  to achieve Label DP. Their ALIBI approach combines Laplace noise and Bayesian inference and improves over the prior work. Moreover, they introduce a memorization attack in the Label DP scenario.


**Limitations And Societal Impact:**

Given the limited algorithmic novelty in PATE-FM and the interesting experiments, I think the paper is marginally above the acceptance threshold.   There isn't potential negative societal impact.

**Main Review:**


Clarity: The paper is well written and easy to follow. The background and related work section is complete and helpful.

Weak points:
1. The proposed PATE-FM lacks algorithmic novelty.  There is no clear difference between PATE and the proposed PATE-FM except that teachers in PATE-FM can jointly reuse unlabeled private data. In other words, the label-DP was supposed to improve performance by sacrificing privacy protections for features. However, the authors use it in a naive way ---  incorporating more unlabeled free data in training. Therefore, I did not see the clear connections between the PATE-FM and label-DP as we can apply the same semi-supervised training in the DP-SGD algorithm and call it an improved algorithm under label-DP.

Strong points:
1. The paper studies a new and important problem --- improving the privacy-utility trade-offs in the label-DP setting.

2. This paper provides an interesting view on comparing different DP approaches. They empirically estimate the privacy loss based on the black-box membership inference. Moreover, the findings on ``PATE-EM is much less likely to memorize labels compared to ALIBI’’ are novel though the constructed attack may not be the strongest possible attack.

Question:
Usually, we consider the Gaussian mechanism is stronger than the Laplace mechanism due to its more concentrated noise. I am confused on why  ALIBI improves over iterative Bayesian inference on additive Gaussian noise?

**Time Spent Reviewing:**

6 hours.

---

> ### Author Response · Authors · 2021-08-10
> **Response to review**
>
> > However, the authors use it in a naive way --- incorporating more unlabeled free data in training.
>
> The synergy between PATE and Label DP, while natural, had not been explored prior to our work. This is despite the fact that the sample-and-aggregate technique (of which PATE is an instance) was introduced in 2007, label-only DP setting in 2011, PATE in 2017, and the MixMatch (an advanced semi-supervised learning algorithm) was evaluated in application to PATE in 2019.
>
> > I did not see the clear connections between the PATE-FM and label-DP as we can apply the same semi-supervised training in the DP-SGD algorithm and call it an improved algorithm under label-DP.
>
> The combination of PATE and semi-supervised training uniquely benefits from the Label DP setting. First, PATE requires unlabeled public data, the assumption that, in practice, is difficult to satisfy outside of Label DP. Second, PATE partitions data among teachers, which, as their number increases, leads to data starvation. Semi-supervised training can be quite efficient in the few-label regime as long as the unlabeled data is plentiful.
>
> DP-SGD can be applied to any semi-supervised learning algorithm such as FixMatch. However, DP-SGD’s design and analysis are predicated on the secrecy of the batch composition, which requires that the labels _and_ the features need to be hidden from the adversary. In other words, DP-SGD achieves features-and-labels DP and as such is an overkill for the Label DP setting.
>
> > I am confused on why ALIBI improves over iterative Bayesian inference on additive Gaussian noise?
>
> The Gaussian noise has sharper tails and better composition properties than Laplace. However, its variance $(~2 \ln(1.25/\delta)/\epsilon^2)$ is _larger_ than Laplace’s $(2/\epsilon^2)$ for the same $\epsilon$ and $\delta < 0.4$. Since ALIBI applies its privacy mechanism only once to each label, its composition behavior is largely irrelevant. (We note that the version of PATE that we use does apply the Gaussian mechanism precisely because of its thinner tails and stronger composition.)
>
> We tested an ALIBI variant instantiated with the Gaussian noise (“AGIBI”). These results follow the same format as Table&nbsp;1 of the submission.
>
> | AGIBI              | Accuracy | $\epsilon$ | 95% CI-$\epsilon_m$ |
> |--------------------|----------|------------|-----------------------|
> | CIFAR-10 High Acc  | 93.5%    | 19         | 2.1&ndash;3.2             |
> | CIFAR-10 Med Acc   | 84.3%    | 7.7        | 0.7&ndash;1.4             |
> | CIFAR-10 Low Acc   | 71.3%    | 3.7        | 0.0&ndash;2.0             |
> | CIFAR-100 High Acc | 69.9%    | 17         | 2.7&ndash;3.4             |
> | CIFAR-100 Med Acc  | 50.8%    | 8.4        | 1.3&ndash;2.0             |
> | CIFAR-100 Low Acc  | 28.7%    | 5.4        | 0.6&ndash;1.1             |
> ||
>
> The empirical bounds on privacy loss for ALIBI and AGIBI substantially overlap, while theoretical bounds strongly favor ALIBI.

---

> > ### Author Response · Authors · 2021-08-19
> > **Have concerns been addressed?**
> >
> > We'd like to follow up on our response, and especially on the new data that we provided. Does it adequately address the reviewer's question about Laplace vs Gaussian noise?

---

> > > ### Comment · Reviewer_ufMh · 2021-08-19
> > > **Reply**
> > >
> > > Thanks for the authors' reply, especially for the question on Gaussian noise.
> > > I apologized for did not make it clear for the first concern on PATE-EM.  Regarding improving DPSGD using semi-supervised training in the label DP setting, I referred to semi-supervised training where the privacy cost only counts for fully labeled data. Therefore, the DPSGD can still benefit from those unlabeled training data without spending an extra privacy budget.  If authors use label-DP only by incorporating more unlabeled free data in teacher training, then I do not think it is a novel DP algorithm because other DP algorithms (e.g., DPSGD) can benefit from the label-DP setting in the same way.

---

> > > > ### Author Response · Authors · 2021-08-20
> > > > **DP-SGD for SSL**
> > > >
> > > > > Therefore, the DPSGD can still benefit from those unlabeled training data without spending an extra privacy budget.
> > > >
> > > > While it is conceivable (and indeed probable) that some SSL techniques combine well with DP-SGD, MixMatch and its derivatives are unlikely candidates for the following reason. MixMatch, as presented in the original [paper](https://arxiv.org/pdf/2001.07685.pdf), balances the losses computed over labeled and unlabeled examples within each batch, regardless of the ratio of labeled to unlabeled data in the training dataset. As such, it will effectively incur DP-SGD-like losses, assuming the same clipping and noise parameters.

---

### Official Review · Reviewer_K4t7 · 2021-07-22

**Rating:** 6
**Confidence:** 4

**Summary:**

The paper studies the problem of supervised learning under label privacy, which only imposes differential privacy constraints on the labels of training samples. The paper proposes two algorithms based on different privatization techniques. (1) PATE-FM: The first algorithm divides samples into $K$ equal subsets and trains $K$ non-private teacher classifiers using FixMatch, a semi-supervised learning technique with the unlabeled features from all $n$ samples and labels of samples from the corresponding subset. Then the algorithm trains a private student network using the PATE framework, which is based on private processing of predictions from teacher classifiers. (2) ALIBI: The algorithm randomly perturbs the label of each sample by adding Laplace noise onto the one-hot encoding of the labels. After normalizing the noisy vector, the private soft labels can be then used for subsequent training.

The paper then performs experiments to evaluate the privacy/accuracy tradeoff for both methods and the LP-2ST algorithm proposed in reference [17]. The paper also performs label inference attacks to estimate the empirical privacy for both methods.

**Limitations And Societal Impact:**

Yes.

**Main Review:**

The paper is nicely written.  Although the paper builds heavily on previous techniques in the literature, the paper still has some interesting ideas, i.e, combining semi-supervised learning techniques with PATE to preserve label privacy. I am tending towards accepting the paper.

Comments:
1. In table one, the privacy utility tradeoff for PATE-FM is better on CIFAR-10 and ALIBI is better on CIFAR-100. Is this related to the size of the dataset (number of labels)?

2. I am not quite convinced by the way the empirical privacy cost is calculated. If 1000 labels are flipped, this might significantly bias the obtained model. Hence more experiments are recommended to support the discussion on empirical losses.




**Time Spent Reviewing:**

4

---

> ### Author Response · Authors · 2021-08-10
> **Response to review**
>
> > In table one, the privacy utility tradeoff for PATE-FM is better on CIFAR-10 and ALIBI is better on CIFAR-100. Is this related to the size of the dataset (number of labels)?
>
> Indeed, provable privacy guarantees favor PATE-FM over ALIBI on the CIFAR-10 dataset, and inverse of that on CIFAR-10. The causes of PATE-FM’s poor provable privacy in the many-classes regime are twofold:
> 1. Fewer teachers that can achieve target performance levels due CIFAR-100’s lower samples-per-class ratio (200-800 teachers for CIFAR-10 vs 20-100 teachers for CIFAR-100, Table 3 in the Appendix).
> 2. More queries required by the student (250&ndash;500 for CIFAR-10 and 1,000&ndash;9,400 for CIFAR-100, _ibid_).
>
> These guarantees can be contrasted with the alternative measurements&mdash;empirical privacy loss&mdash;that are consistent with PATE-FM’s being more private than ALIBI. Given how close these empirical measurements track ALIBI’s theoretical bounds, we conjecture that PATE-FM’s analysis can be strengthened.
>
> > If 1000 labels are flipped, this might significantly bias the obtained model. Hence more experiments are recommended to support the discussion on empirical losses.
>
> Due to relative inefficiency of PATE-FM, we focus on demonstrating that 1,000 flipped labels do not substantially change our conclusions about memorization of ALIBI. For each target accuracy level, we trained 10 models on datasets with 100 flipped labels each, and applied the same memorization test as before.
>
> The table below demonstrates that the confidence intervals for the original attack (1,000 flipped labels) and the 10&times;100 attack are extremely close.
>
> |                    | ALIBI (1000 labels) | ALIBI (10&times;100 labels) |
> |--------------------|---------------------|-----------------------|
> | CIFAR-10 High Acc  | 2.7&ndash;4.0             | 2.7&ndash;3.6               |
> | CIFAR-10 Med Acc   | 0.3&ndash;2.2             | 0.2&ndash;2.5               |
> | CIFAR-10 Low Acc   | 0.0&ndash;0.4             | 0.0&ndash;1.6               |
> | CIFAR-100 High Acc | 2.8&ndash;3.7             | 2.7&ndash;3.4               |
> | CIFAR-100 Med Acc  | 1.4&ndash;2.4             | 1.3&ndash;2.0               |
> | CIFAR-100 Low Acc  | 0.5&ndash;1.0             | 0.6&ndash;1.1               |
> ||
>
> Since the success rate of memorization inference does not change as the number of flipped labels changes by a factor of 10, it offers strong evidence that our heuristic methodology is valid.

---

> > ### Author Response · Authors · 2021-08-19
> > **Have concerns been addressed?**
> >
> > We'd like to follow up on our response, and especially on the new data that we provided. Does it adequately address the reviewer's concern about the bias possibly introduced by the flipped labels?

---

> > > ### Comment · Reviewer_K4t7 · 2021-08-19
> > > **Reply**
> > >
> > > Thanks for the reply. They have addressed my questions. It would be nice the new experiment results can be included or discussed in future updates.

---

> > > > ### Author Response · Authors · 2021-08-27
> > > > **Reply**
> > > >
> > > > Thanks for the confirmation. We will include a summarized version in the main text and we will update the supplementary materials with more details.

---

### Author Response · Authors · 2021-08-27
**Pending questions**

We would like to thank the reviewers for their questions and comments. The paper has definitely improved as a result. We would like to check one last time if there are any pending questions that we have not adequately addressed.

---

### Decision · Program_Chairs · 2021-09-27

**Decision:**

Accept (Poster)

**Comment:**

In private deliberation, reviewers seemed to feel that the paper was sound, but perhaps not the most exciting (in particular, not the most novelty in PATE-FM). Nonetheless, they felt it was thorough enough and above the bar for NeurIPS.